# Dendritic Polyglycerol Sulfate Reduces Inflammation Through Inhibition of the HMGB1/RAGE Axis in RAW 264.7 Macrophages

**DOI:** 10.3390/ijms262110440

**Published:** 2025-10-27

**Authors:** Marten Kagelmacher, Cristina S. Quella, Emma Kautz, Anna Klumpp, Felix Weichert, Issan Zhang, Dusica Maysinger, Poornima G. Wedamulla, Suzana K. Straus, Thomas Risse, Rainer Haag, Marina Pigaleva, Jens Dernedde

**Affiliations:** 1Institute of Chemistry and Biochemistry, Freie Universität Berlin, 14195 Berlin, Germany; m.kagelmacher@fu-berlin.de (M.K.);; 2Institute for Laboratory Medicine, Clinical Chemistry, and Pathobiochemistry, Charité University Medicine Berlin, 13353 Berlin, Germany; 3Department of Pharmacology and Therapeutics, McGill University, Montreal, QC H3A 0G4, Canada; 4Department of Chemistry, University of British Columbia, Vancouver, BC V6T 1Z1, Canada

**Keywords:** high mobility group box 1, receptor for advanced glycation end products, dendritic polyglycerol sulfate, macrophages, inflammation, cytokines

## Abstract

High Mobility Group Box 1 (HMGB1) is a central pro-inflammatory mediator released from damaged or stressed cells, where it activates receptors such as the Receptor for Advanced Glycation Endproducts (RAGE). Dendritic polyglycerol sulfate (dPGS), a hyperbranched polyanionic polymer, is known for its anti-inflammatory activity. In this study, we examined how dPGS modulates HMGB1-driven signaling in RAW 264.7 macrophages and human microglia. Recombinant human HMGB1 expressed in *Escherichia coli* (*E. coli*) was purified by nickel-nitrilotriacetic acid (Ni-NTA) and heparin chromatography. Proximity ligation assays (PLA) revealed that dPGS significantly disrupted HMGB1/RAGE interactions, particularly under lipopolysaccharide (LPS) stimulation, thereby reducing inflammatory signaling complex formation. This correlated with reduced activation of the nuclear factor kappa B (NF-κB) pathway, demonstrated by decreased nuclear translocation and transcriptional activity. Reverse transcription polymerase chain reaction (RT-PCR) and quantitative real-time PCR (RT-qPCR) showed that dPGS suppressed HMGB1- and LPS-induced transcription of tumor necrosis factor alpha (TNF-α), interleukin-6 (IL-6), monocyte chemoattractant protein-1 (MCP-1), cyclooxygenase-2 (COX-2), and inducible nitric oxide synthase (iNOS). Enzyme-linked immunosorbent assay (ELISA) and Griess assays confirmed reduced TNF-α secretion and nitric oxide production. Electron paramagnetic resonance (EPR) spectroscopy further showed that dPGS altered HMGB1/soluble RAGE (sRAGE) complex dynamics, providing mechanistic insight into its receptor-disruptive action.

## 1. Introduction

Inflammation is a complex biological response to harmful triggers such as pathogens, damaged cells, or irritants, and is a crucial component of the innate immune system. However, excessive or chronic inflammation contributes to the pathogenesis of various diseases, including sepsis, rheumatoid arthritis, and atherosclerosis [1]. One of the key mediators of chronic inflammation is the ubiquitously expressed High Mobility Group Box 1 (HMGB1) protein, a DNA-binding, non-histone-like nuclear protein involved in nucleosome sliding, transcription enhancement, DNA repair, and telomere homeostasis [2,3,4]. When HMGB1 is released extracellularly due to responses to cellular stress or injury, it functions as a damage-associated molecular pattern (DAMP), or alarmin [5]. HMGB1 exerts its pro-inflammatory effects primarily through interaction with pathogenic molecules, like lipopolysaccharide (LPS), bacterial or viral DNA/RNA, and via the receptor for advanced glycation end products (RAGE), activating downstream signaling pathways that amplify inflammatory responses [6]. RAGE is a transmembrane pattern recognition receptor (PRR) belonging to the immunoglobulin superfamily and is widely expressed on immune cells, endothelial cells, and epithelial tissues. It is a multiligand receptor that interacts with a diverse range of endogenous and exogenous ligands, including HMGB1, S100 proteins, amyloid-ß and advanced glycation end products (AGEs) [7,8,9]. These interactions play a pivotal role in inflammation, neurodegeneration, diabetes, and cancer. Upon recognition of its ligands, RAGE oligomerizes on the plasma membrane to enhance ligand affinity and promotes effective activation of intracellular signaling cascades [10]. This oligomerization is essential for initiating mitogen-activated protein kinases (MAPKs), Janus kinase/signal transducers and activators of transcription (JAK/STAT), phosphoinositide 3-kinase (PI3K)/Akt pathways, and nuclear factor kappa B (NF-κB) translocation into the nucleus [11]. Activation of these signaling pathways results in the upregulation of pro-inflammatory cytokines and chemokines such as tumor necrosis factor-alpha (TNF-α), interleukin-6 (IL-6), monocyte chemoattractant protein-1 (MCP-1), and interleukin-1 beta (IL-1β), which further perpetuate the inflammatory response [12,13]. The chronic activation of RAGE signaling has been implicated in various pathological conditions, making it a key therapeutic target. Structurally, HMGB1 is a 25 kDa non-histone chromatin-associated protein composed of two DNA-binding HMG box domains (A- and B-box) and a negatively charged C-terminal tail. The A- and B-box of HMGB1 each contain three alpha-helices that allow for DNA binding and interactions with external molecular triggers, such as LPS, bacterial and viral DNA/RNA. The B-box has been identified as the primary domain responsible for the pro-inflammatory activity of HMGB1, specifically in its interaction with RAGE [14,15]. The HMGB1/RAGE binding interface is largely facilitated by electrostatic interactions, particularly between the acidic tail of HMGB1 and the positively charged ligand-binding domain of RAGE [16]. HMGB1 contains three redox-sensitive cysteine residues (Cys23, Cys45, and Cys106), which are crucial in regulating its activity. Cys23 and Cys45, located in the A-box, form an intramolecular disulfide bridge under oxidative conditions, while Cys106, found in the B-box, remains in a thiol form and plays a key role in its pro-inflammatory activity. The oxidation state of these cysteines determines HMGB1 functional properties: in its reduced form, HMGB1 promotes chemotaxis through C-X-C chemokine receptor 4 (CXCR4), while the disulfide form is responsible for pro-inflammatory cytokine induction via RAGE and Toll-like receptor 4 (TLR4) [17,18]. Complete oxidation of HMGB1 to a sulfonyl form abolishes its inflammatory activity [19]. A nuclear magnetic resonance (NMR) spectroscopy study has provided insights into the structural dynamics of HMGB1, revealing that its redox state influences its pro-inflammatory activity and heparin-binding affinity [20]. Following ligand binding, RAGE undergoes internalization, a process that is crucial for regulating receptor-mediated signaling. Internalized RAGE-ligand complexes traffic into endosomes and subsequently to lysosomes, where they are either degraded or contribute to sustained intracellular signaling [21,22]. This endocytic pathway is particularly important in macrophages, where internalized HMGB1/RAGE complexes can perpetuate inflammatory responses even after ligand removal from the extracellular space. Therefore, the modulation of RAGE internalization presents an additional therapeutic avenue in inflammation-associated pathologies. Given its central role in inflammation and immune regulation, the HMGB1/RAGE axis has emerged as a promising target in various diseases. In sepsis, HMGB1 is released as a late mediator of inflammation, exacerbating systemic inflammation and contributing to multiple organ failure [23]. Additionally, chronic RAGE activation is implicated in neurodegenerative diseases such as Alzheimer’s disease, where the receptor binds amyloid-β and facilitates neuronal inflammation and oxidative stress [24]. Targeting HMGB1/RAGE signaling has the potential to mitigate excessive inflammatory responses and restore immune homeostasis. Several strategies have been explored. These include small-molecule inhibitors, neutralizing antibodies, and competitive ligands that prevent HMGB1 from binding to RAGE. Heparin, a naturally occurring polyanion, is well known for its anti-inflammatory properties and can inhibit the interaction of HMGB1 with RAGE, thereby reducing downstream inflammatory signaling [25]. Similarly, dendritic polyglycerol sulfate (dPGS) has emerged as a potential anti-inflammatory agent due to its ability to interact with key inflammatory mediators. Originally developed as a synthetic polyanion with a well-defined molecular structure, dPGS has been extensively studied for its systemic effects in various inflammatory models. Early studies demonstrated its strong binding affinity to pro-inflammatory proteins such as HMGB1 and its capacity to modulate immune cell responses [26,27]. Additionally, dPGS has been shown to reduce inflammation in experimental models of arthritis, lung injury, and sepsis, further highlighting its therapeutic potential [28]. Macrophages play a central role in inflammation, particularly in the context of HMGB1-mediated immune responses [29]. As key innate immune cells, macrophages recognize and respond to HMGB1 through RAGE, leading to the secretion of pro-inflammatory cytokines that drive tissue damage and systemic inflammation. In sepsis, activated macrophages contribute to a cytokine storm, exacerbating disease severity [30]. The modulation of macrophage activity by inhibiting HMGB1 signaling represents a potential therapeutic approach to control hyperinflammation states. Recent studies suggest that compounds like dPGS can inhibit macrophage activation by blocking HMGB1/RAGE interactions, thereby reducing inflammatory cytokine production and improving outcomes in inflammatory diseases. In this study, we recombinantly expressed and purified human HMGB1 and applied it as an alarmin to stimulate RAW 264.7 macrophages. Using proximity ligation assays (PLA), we first confirmed that dPGS significantly disrupts the interaction between HMGB1 and RAGE in both mouse macrophages and human microglia, pointing to an extracellular mechanism of action. Subsequent immunocytochemical analyses demonstrated that HMGB1 stimulation leads to nuclear translocation of NF-κB in macrophages, a process markedly attenuated by co-treatment with dPGS. Gene expression analyses via RT-PCR and RT-qPCR revealed that dPGS strongly reduces the mRNA levels of key inflammatory cytokines and mediators, including TNF-α, IL-6, MCP-1, COX-2, and iNOS, in both HMGB1- and LPS-stimulated cells. At the protein level, co-treatment with dPGS suppressed TNF-α secretion and showed reduced nitric oxide (NO) release. Further, flow cytometry uptake studies demonstrated time-dependent internalization of red fluorescently cyanin 5 (Cy5) labeled dPGS into macrophages, with altered uptake dynamics upon co-stimulation with HMGB1 or LPS. Finally, electron paramagnetic resonance (EPR) spectroscopy supported direct interactions between dPGS and the HMGB1/soluble (s)RAGE complex, suggesting that dPGS alters the conformation and assembly of this inflammatory axis. Taken together, these findings demonstrate that dPGS inhibits the HMGB1/RAGE signaling at multiple levels, extracellular binding, intracellular signaling, and receptor-ligand complex assembly, underscoring its potential as a versatile therapeutic agent to diminish inflammatory processes driven by HMGB1 or LPS.

## 2. Results

### 2.1. Cloning, Recombinant Expression, and Purification of Human HMGB1 from E.coli

The HMGB1 protein was successfully cloned into a modified pET-22b(+) expression vector (see Appendix A). Expression of HMGB1 was performed in the *E. coli* strain BL21(DE3). Successful expression in the soluble fraction can be seen in Appendix A in the lane + IPTG, where a distinct band at around 26 kDa is observed. This is in line with the theoretical molecular weight of recombinant HMGB1. A two-step purification process on Ni-NTA agarose, followed by heparin Sepharose chromatography, was applied to obtain highly pure HMGB1. The chromatogram of the Ni-NTA purification can be seen in Appendix A. The protein of interest eluted in fractions 17–28, which corresponds to a Mol% of imidazole of 22 to 80 Mol% (or 120 to 400 mM imidazole) peaking between 49 to 55 Mol% (or 250 to 280 mM imidazole) (compare Appendix A). Notably, a second band at around 20 kDa was observed, which has also been reported in earlier studies as a potential degradation product or truncated form of HMGB1 (compare Appendix A) [31,32]. Leveraging the intrinsic heparin-binding affinity of HMGB1, attributed to the A-box, the protein was further purified on heparin Sepharose. The Ni-NTA purified and concentrated protein suspension was loaded onto heparin Sepharose, and proteins that did not bind to heparin can be seen in fractions 5–6 (see Appendix A). The main protein peak that was eluted from the heparin Sepharose column eluted in fractions 28–30 at a salt concentration between 22 to 53 Mol% (or 430 mM to 1060 mM NaCl). SDS-PAGE analysis revealed that HMGB1 eluted between 749 mM and 1061 mM NaCl. Due to the co-elution of the second band at 38 Mol% (750 mM NaCl), fractions from 43 to 53 Mol% (or 850 mM till 1061 mM) were collected and further concentrated (see Appendix A). The final purified HMGB1 exhibited an A_260nm/280nm_ ratio of 0.6, generally accepted to be a sign of high purity with minimal nucleic acid contamination. Protein yield was quantified at 4.95 mg per liter of bacterial culture.

### 2.2. HMGB1/RAGE Interaction Is Reduced by dPGS

HMGB1 is a nuclear protein that is released from stimulated macrophages and acts as an alarmin. Recognition of extracellular HMGB1 is realized by the transmembrane receptor RAGE, which propagates inflammatory responses. In macrophages, the HMGB1/RAGE axis plays a crucial role in mediating immune activation, particularly in response to bacterial endotoxins such as LPS [33,34]. As it is known that the polyanion dPGS, as well as heparin, binds to basic amino acid patches of proteins [26,35], it was hypothesized that dPGS treatment may alter HMGB1/RAGE interaction. Here, we applied the PLA as a suitable method in a cellular context. The results of the PLA targeting HMGB1/RAGE complexes in RAW 264.7 macrophages are presented in Figure 1A,B. Under control (vehicle only) conditions, complexes of HMGB1/RAGE are detectable, suggesting a basal level expression of both proteins, considering that the depletion of serum in the culture medium potentially induces minimal cellular stress reactions (Figure 1A). Upon stimulation with 100 ng/mL LPS, a notable increase in the PLA signal is observed, indicative of an enhanced HMGB1/RAGE interaction. This effect is accompanied by a change in the cytoskeleton, shown by an increase in filopodia and lamellipodia, which are thin, actin-rich protrusions and broader, sheet-like projections, respectively, as shown by the phalloidin staining. Such structures are a result of cell activation, enabling macrophages to interact with their environment and enhance phagocytosis [36]. Although treatment with 50 nM dPGS alone did not significantly affect the PLA signal compared to the control, co-treatment of LPS-stimulated cells with dPGS resulted in a significant reduction in the PLA signal, indicating that dPGS attenuates LPS-induced activation. Also, the LPS-induced change in macrophage morphology was slightly rectified, suggesting that dPGS exerts a cell-protective effect. We observed similar effects in human microglia (see Appendix A), where a basal level of HMGB1/RAGE interaction is detectable, indicating low-level constitutive interaction or stress-induced basal signaling, but upon addition of LPS, a significant increase in HMGB1/RAGE signal can be seen. This also suggests enhanced pro-inflammatory signaling via HMGB1/RAGE in response to LPS exposure. Again, treatment with dPGS alone did not significantly affect basal HMGB1/RAGE interactions but maintained the level of untreated control. Similarly to the macrophages, results in microglia showed a significant reduction in PLA signals upon co-treatment with LPS and dPGS, indicating that dPGS’s ability to effectively counteract LPS-mediated enhancement of HMGB1/RAGE interactions is found in different cell types. The effect is likely through competitive inhibition at the receptor level or by disrupting downstream signaling pathways.

### 2.3. Shuttling of NF-κB in HMGB1 and LPS-Stimulated Macrophages Is Reduced by dPGS

Following up on our results showing that dPGS effectively inhibits the complexation of HMGB1/RAGE on mouse macrophages, we next focused on the inflammatory signaling route by studying the effect of dPGS on the translocation of the transcription factor NF-κB into the nucleus upon LPS stimulation. In general, the NF-κB transcription factor is one of the key regulators of inflammatory responses. In the absence of a stimulus (“resting” conditions), NF-κB is sequestered in the cytoplasm by inhibitor protein (IκB). Upon stimulation with pro-inflammatory stimuli such as LPS or HMGB1, signaling cascades lead to the degradation of IκB, allowing NF-κB to translocate into the nucleus, where it drives the transcription of inflammatory genes [33,37]. This pathway plays a central role in macrophage activation and the release of cytokines during infection and inflammation. This well-characterized signaling mechanism provided the basis for investigating whether NF-κB activation and nuclear translocation are affected by the treatment with pro-inflammatory stimuli and potential anti-inflammatory agents. To explore this, the intracellular localization of NF-κB in RAW 264.7 macrophages was examined using immunofluorescence microscopy and quantified based on the MFI within the nuclear region (see Figure 2A,B). Under control conditions, there was a minimal NF-κB signal in the nucleus, consistent with the inactive state of resting macrophages. As expected, stimulation with the DAMP HMGB1 or LPS significantly increased the NF-κB fluorescence signal within the nucleus (see Figure 2B), while treatment with dPGS alone did not lead to a substantial increase in NF-κB nuclear translocation. Interestingly, co-treatment of dPGS together with HMGB1 or LPS reduced the nuclear NF-κB signal comparable to the negative control, suggesting an inhibitory effect on the NF-κB activation pathway, which is consistent with the reduction in HMGB1/RAGE complexes observed previously in the PLA results (Figure 1). Also, a direct interaction of dPGS with cytoplasmic NF-κB can be assumed to diminish nuclear transport.

### 2.4. Dendritic Polyglycerol Sulfate Reduces mRNA Levels of Cytokines

As we showed, dPGS reduces the amount of nuclear NF-κB, which adds to the anti-inflammatory character of dPGS. The next step was to investigate whether mRNA levels of cytokines and chemokines were also affected by co-stimulation of mouse macrophages with HMGB1 or LPS and dPGS. Upon activation, macrophages respond with transcriptional upregulation of pro-inflammatory proteins due to the translocation of transcription factors such as NF-κB. Inflammatory mediators that play important roles are TNF-α, IL-6, MCP-1, COX-2, and iNOS. LPS and HMGB1 are well-known triggers of macrophage signaling pathways, including the Toll-like receptor (TLR) and RAGE pathways, which regulate cytokine and chemokine gene expression [38,39]. However, their effects on specific pro-inflammatory gene expression are distinct based on receptor interactions and downstream signaling cascades. Therefore, we analyzed here the influence of dPGS on transcriptional activity. The RT-PCR results in Figure 3A–D show that LPS robustly upregulates MCP-1, IL-6, and TNF-α mRNA levels, with significant fold changes relative to the untreated control. This reflects the broad activation of NF-κB and other inflammatory transcription factors by LPS. On the other hand, HMGB1 selectively induces TNF-α mRNA upregulation, while MCP-1 and IL-6 levels remain close to the control level. A possible explanation for this selective response is the differential activation of signaling pathways by HMGB1 compared to LPS. While LPS activates TLR4 signaling through MyD88 and TRIF, leading to NF-κB and interferon regulatory factor (IRF) activation, HMGB1 primarily signals through RAGE and TLR4 but with a preference for NF-κB-dependent TNF-α transcription, rather than broad inflammatory activation [40,41]. Additionally, HMGB1-induced cytokine expression may require synergy with other DAMPs or secondary mediators, which could be absent in the current experimental setting, limiting MCP-1 and IL-6 upregulation [17]. Treatment with 50 nM dPGS alone does not significantly alter MCP-1 and TNF-α expression but reduces the mRNA levels of IL-6, as expected by the fact that dPGS does not induce pro-inflammatory stimuli. However, in co-treatment experiments together with HMGB1 or LPS, dPGS significantly reduces the mRNA levels of TNF-α, IL-6, and MCP-1, indicating that dPGS exerts an anti-inflammatory effect by interfering with proteins of the signaling cascade. Additional RT-qPCR results (see Appendix A) focus on the LPS-induced cytokine expression and the modulatory effects of dPGS and polymyxin B (PMB). Again, LPS strongly induces MCP-1, IL-6, TNF-α, COX-2, and iNOS mRNA levels. PMB, a known LPS-neutralizing agent, significantly reduced LPS-induced cytokine expression across all targets. Like PMB, dPGS markedly suppressed LPS-induced cytokine expression.

### 2.5. Dendritic Polyglycerol Sulfate Reduces the Release of the Protein TNF-α and NO in HMGB1 and LPS-Stimulated Mouse Macrophages

Since dPGS also effectively reduced the mRNA levels of the respective cytokines, we next investigated whether this effect can also be translated to the protein and nitric oxide (NO) levels. Therefore, we co-stimulated RAW 264.7 macrophages with HMGB1 or LPS in increasing concentrations of dPGS under serum-free conditions. Both stimuli induced a significant production of TNF-α compared to the untreated control, whereas LPS elevated the TNF-α levels even further (increase by 4-fold compared to HMGB1 alone, see Figure 4A,B). However, co-treatment of HMGB1 with dPGS results in a significant decrease in TNF-α, already reaching a maximal inhibition level of around 50% at 0.05 nM dPGS. Interestingly, increased concentrations of the polymer did not result in a dose-dependent inhibition. Co-treatment of LPS with 50 nM dPGS also showed a significant inhibition (Figure 4B). The polymer control (500 nM) showed no significant production of TNF-α compared to the untreated control. In addition to measuring cytokine levels, the release of nitric oxide (NO) is also critical to follow inflammatory responses, which is predominantly induced through the nitric oxide synthase (iNOS) pathway. Here, the effects of dPGS on NO production in LPS-stimulated RAW 264.7 macrophages were investigated. As shown in Appendix A, LPS stimulation alone induced a robust increase in NO-production. Co-treatments with increasing concentrations of dPGS resulted in a marked reduction in NO-release at micromolar concentrations, while at the highest concentration, NO release could not be observed. This shows that dPGS can act as an anti-inflammatory compound in the context of HMGB1 and LPS-stimulated RAW 264.7 macrophages, which requires other mechanisms than interaction with PRRs, DAMPs, and alarmins. We further showed that all treatments were not cytotoxic by performing a mitochondrial activity assay (MTT) (see Appendix A).

### 2.6. Uptake of dPGS in Mouse Macrophages Is Altered When Co-Stimulated with HMGB1 and LPS

At this point, we showed that dPGS affected multiple inflammatory routes outside and inside the cell. Therefore, the cellular uptake of dPGS is a prerequisite to target intracellular proteins and mediate anti-inflammatory effects. In this context, we fluorescently labeled dPGS and incubated it with RAW 264.7 macrophages for varying durations, and the uptake was assessed by flow cytometry. As shown in Appendix A, dPGS-Cy5 intracellular accumulation increased over time, with peak MFI observed at 16–24 h, where a plateau is reached. This suggests that macrophages internalize dPGS steadily over time. The absence of serum in the medium likely favored internalization by reducing competitive binding to serum components. This finding confirms that dPGS-Cy5 is efficiently internalized by macrophages, potentially enabling intracellular interactions with the inflammatory signaling pathway. In Figure 5, the effects of co-incubation of the macrophages with dPGS-Cy5 (50 nM) and either HMGB1 (1 µg/mL) or LPS (100 ng/mL) on the level of dPS-Cy5 uptake were investigated for 1, 6, 16, and 24 h. The flow cytometry results revealed distinct temporal patterns of dPGS-Cy5 uptake modulation depending on the inflammatory stimulus. Co-treatment with HMGB1 already showed a significant reduction in dPGS-Cy5 uptake after 1 h, indicating rapid effects of HMGB1 on polymer internalization. This reduction persisted across all time points (6, 16, 24 h), suggesting that HMGB1 may either compete with dPGS for receptor-mediated binding or alter the endocytic pathways, potentially by stabilizing the cell membrane or modifying surface receptor availability. In contrast, co-treatment with LPS did not significantly reduce dPGS-Cy5 uptake at the 1 h mark but showed a gradual decrease from the 6 h time point onwards, which became more pronounced at 16 and 24 h. In fact, for 6 h and all later time points, the effect was indistinguishable from the HMGB1 effect. This suggests that LPS or HMGB1-induced cellular changes, such as cytoskeletal rearrangement or membrane stiffening associated with macrophage activation, may progressively inhibit dPGS internalization over time. The delayed onset of reduced uptake with LPS compared to HMGB1 highlights distinct cellular responses to PAMPs versus DAMPs, with LPS possibly inducing longer-term effects on endocytic processes.

### 2.7. Spectroscopic Evidence for dPGS Interaction with HMGB1/sRAGE Complexes

To gain further insights at a molecular level, the interaction between HMGB1, sRAGE, and dPGS was investigated with electron paramagnetic resonance (EPR) spectroscopy. To this end, a S-(1-oxyl-2,2,5,5-tetramethyl-2,5-dihydro-1H-pyrrol-3-yl)methyl methanesulfonothioate label (MTSL) was covalently tethered to the naturally occurring Cys106 residue located in the first α-helix of the B-box domain of HMGB1. This site is situated within the region of the protein that is reported to be involved in sRAGE binding [42]. The recorded EPR spectrum of spin-labeled HMGB1 displayed an anisotropic line shape with well-defined, narrow low- and high-field peaks and a 2A_zz_ value of 61 G (Figure 6A, black trace). The spectrum is consistent with a solvent-exposed helical site expected based on the structure of HMGB1 [43]. Upon mixing spin-labeled HMGB1 with sRAGE, the spectrum (Figure 6A, red trace) shows a between the outer extrema, the so-called 2A_zz_-value of 62.5 G increased by 1.5 G as compared to HMGB1 (black trace, 2A_zz_ = 61 G). The increased 2A_zz_ value is consistent with the formation of HMGB1/sRAGE complexes, as the latter exhibits a smaller global rotational diffusion rate. Additionally, the changes in the line shape are most prominently seen on the high field side of the central minimum (see arrows). Such changes are typically observed if there are additional changes in the local mobility of the paramagnetically labeled side chain. Therefore, the changes in the EPR line shape provide direct evidence for a close involvement of the B-box region in the interaction between HMGB1 and sRAGE. This finding is consistent with previous work, in which the four HMGB1 residues (E145, K146, E153, and E156) located within the B-box domain of HMGB1 were found to interact with RAGE [44]. To elucidate the potential ability of a polyelectrolyte to disrupt the HMGB1/sRAGE complex, the interaction of the pre-formed HMGB1/sRAGE complex with the dPGS was studied. The addition of dPGS to the HMGB/sRAGE sample (see Figure 6A, blue line) resulted in a significant further increase in the value of the 2A_zz_-splitting from 62.5 G to 65.5 G. This strong increase indicates a significant further slowing of the global rotational motion and therefore confirms an interaction between the dPGS and the spin labeled HMGB1 protein. Moreover, a significant additional change in the line shape was detected (see Figure 6A, blue arrow), which suggests considerable further restrictions of the spin label mobility as a consequence of dPGS binding. Given that both HMGB1/sRAGE as well as HMGB1/dPGS interactions are expected to be primarily electrostatically driven, one might expect these to be mutually exclusive. However, the observed spectral changes indicate that HMGB1 can still engage in interaction with dPGS in its sRAGE bound state. While interaction of dPGS with pre-formed HMGB1/sRAGE complex is shown to take place, two possible scenarios need to be discriminated (see scheme on Figure 6B): (1) dPGS may bind competitively to HMGB1, displacing sRAGE and occupying similar or overlapping binding sites within the B-box, or (2) dPGS may form a ternary complex with both HMGB1 and sRAGE. To distinguish between these possibilities, the interaction between HMGB1 and dPGS without sRAGE in the solution was investigated. The resulting EPR spectrum is shown in Figure 6A (green trace). The 2A_zz_ value rises from 61 G (HMGB1 alone) to 64 G (HMGB1 with dPGS), which is 1.5 G smaller than the value observed for dPGS interacting with pre-formed HMGB1/RAGE complex. This is the clear indication that the second proposed scenario is more likely, i.e., dPGS forms a ternary complex with pre-formed HMGB1/sRAGE complex in solution. Moreover, a comparable change in the local feature at 3500 G can be detected again after the addition of dPGS. On the one hand, this feature provides evidence that the interaction of HMGB1 with dPGS is not identical to its interaction with the HMGB1/sRAGE complex and thus further supports the ternary complex scenario. Additionally, this spectral feature can be considered a characteristic and unique signature of the HMGB1/dPGS interaction in this ternary system, as it emerges to this extent only upon the addition of dPGS.

## 3. Discussion

HMGB1 is an archetypal DAMP that drives inflammation by engaging PRRs. HMGB1 binding to RAGE on immune cells triggers NF-κB activation and a cascade of pro-inflammatory gene expression [45,46]. The molecular basis of this interaction was first demonstrated by Hori et al., who identified RAGE as a cellular binding partner for amphoterin (HMGB1) using ligand blotting and affinity chromatography assays. They further confirmed the binding through immunohistochemistry and functional neurite outgrowth assays, establishing that the HMGB1/RAGE interaction mediates both cell signaling and neuronal differentiation [16]. Subsequently, Park et al. employed co-immunoprecipitation and cell-based binding assays to show that HMGB1 can also interact with multiple Toll-like receptors (TLR2, TLR4, TLR9) in addition to RAGE, thereby amplifying pro-inflammatory signaling in immune cells [47]. Together, these foundational studies provided direct biochemical and cellular evidence that RAGE serves as a bona fide receptor for HMGB1 and that this interaction represents a key axis in inflammatory and innate immune responses. In peripheral and central macrophages (microglia), this signaling leads to robust production of cytokines, chemokines, and enzymes such as TNF-α, IL-6, MCP-1, COX-2, and iNOS [45,46]. Recent studies have further confirmed the centrality of this axis. Fan et al. demonstrated that HMGB1 released from astrocytes activates RAGE-dependent NF-κB signaling in microglia, promoting M1 polarization and worsening spinal cord injury, while RAGE inhibition ameliorated inflammatory damage [48]. Similar, Zhou et al. showed that recombinant HMGB1 synergizes with LPS to activate TRAF6-NF-κB signaling in RAW 264.7 cells, increasing IL-6, iNOS, and TNF-α levels, whereas HMGB1 silencing reversed this pro-inflammatory phenotype [49]. Our findings align with this paradigm: recombinant HMGB1 stimulation of RAW 264.7 macrophages markedly induced these NF-κB target genes and cytokines, modeling the sustained inflammatory response observed in systemic inflammation and sepsis. HMGB1 is indeed a well-known late mediator of sepsis that maintains cytokine release and immune cell recruitment even after initial triggers (e.g., endotoxin) subside [23,46]. Elevated extracellular HMGB1 correlates with disease severity in sepsis, and importantly, blocking HMGB1 or its receptors has shown protective effects in preclinical models. For example, neutralizing antibodies or the HMGB1-binding inhibitor glycyrrhizin can rescue mice from lethal endotoxemia [45,50]. Likewise, genetic or pharmacological intervention of RAGE signaling dampens NF-κB-driven inflammation and improves survival in experimental sepsis [51]. These studies underscore that disrupting the HMGB1/RAGE axis is a viable strategy to curb systemic inflammation. Our results demonstrate that dPGS achieves this disruption effectively, with broad implications for inflammatory diseases. Beyond sepsis, HMGB1/RAGE signaling contributes to pathology in sterile inflammation and chronic diseases. In the central nervous system (CNS), HMGB1 released by stressed or dying cells can activate microglia and compromise the blood-brain barrier [52]. To conclude, HMGB1 is implicated in pathologies ranging from traumatic brain injury to neurodegenerative disease and is likely involved in sepsis-associated encephalopathy [46]. In malignant brain tumors derived from glial cells (glioma), the HMGB1/RAGE axis is of particular interest, as it co-opts inflammatory pathways. HMGB1 is often overexpressed in glioma, and its interaction with RAGE on tumor-associated microglia can promote an immune-suppressive, tumor-promoting microenvironment [53]. Activation of RAGE in glioma-infiltrating myeloid cells stimulates MAPK and NF-κB pathways. Upregulation of cytokines, chemokines, and enzymes (e.g., Matrix Metalloproteinase-9 (MMP9)) then facilitates tumor cell migration, invasion, and immune evasion [53]. Our recent work showed that co-treatments of glioblastoma cells with the unsaturated fatty acid docosahexaenoic acid (DHA) and dPGS, as well as aminated dPG (dPGA), showed additive positive effects in killing glioma cells due to disruption of the HMGB1/RAGE axis [54]. Thus, addressing HMGB1/RAGE interaction in immune cells such as macrophages and microglia is beneficial to reduce harmful inflammation and impede tumor progression. Further, we demonstrated that the polyanion dPGS affects HMGB1 or LPS-driven inflammation in immune cells. Co-treatment with dPGS significantly reduced the induced upregulation of pro-inflammatory cytokines (TNF-α, IL-6, MCP-1), as well as iNOS and COX-2, in RAW 264.7 macrophages. This was evidenced by the mRNA level of NF-κB target genes, by decreased protein level for cytokines and reduced NO production of iNOS. We also showed that dPGS effectively reduced the shuttling of NF-κB p65 into the nucleus upon stimulation with LPS and HMGB1 in macrophages. In earlier work, we reported that dPGS directly binds to HMGB1 and induces oxidation of its three redox-active cysteines (Cys23, Cys45, Cys106), converting it into a non-pro-inflammatory form. In mixed neuron-glia cultures, dPGS prevented LPS-induced dendritic spine loss by neutralizing HMGB1-mediated neuroinflammation [26]. Likewise, Wu et al. showed that dPGS modulates macrophage metabolism by shifting cells toward glycolytic activity, transiently increasing MCP-1 but ultimately exerting net anti-inflammatory effects under stress conditions [55]. Targeting NF-κB by small-molecule inhibitors has been described. For example, a synthetic molecule, BAY 11-7082, and the natural compound parthenolide act inside the cell by directly targeting the NF-κB pathway enzymes. As a result, the inhibitor complex is stabilized in the cytosol, and NF-κB is retained in the cytoplasm [56,57]. We propose a similar mode of action for the polyanion dPGS. Importantly, our data suggest that dPGS acts at both extracellular and intracellular levels. Extracellularly, dPGS serves as a decoy that sequesters HMGB1. EPR analyses indicate direct binding of dPGS to HMGB1/sRAGE complexes in solution. This implies that dPGS can simultaneously engage HMGB1 and the RAGE receptor (here in a soluble decoy form), possibly forming ternary complexes. Such multivalent binding is feasible given the highly sulfated, dendritic structure of dPGS. Indeed, the charged surface of the HMGB1/RAGE complex provides multiple binding sites for polyanions like dPGS. Structural modeling showed that the negatively charged tail of HMGB1 is important to facilitate binding to RAGE by a positively charged pocket [54]. While our data do not provide evidence for a competitive binding to sRAGE of dPGS with HMGB1/sRAGE complexes, it may still impact the immunological response. dPGS may properly alter the membrane association of these complexes, which is important for the proper transmission of the signal. Flow cytometry showed that macrophages readily internalize labeled dPGS over time. Intracellular accumulation of dPGS is consistent with other reports that show that dPGS can enter phagocytic cells and even penetrate 3D brain organoids [58]. Once inside endosomal compartments, dPGS could influence key inflammatory pathways. For example, RAGE-mediated endocytosis of HMGB1 can carry co-ligands like LPS into endolysosomes, triggering inflammasome activation and pyroptosis [59]. Notably, other nanomaterials have shown the ability to attenuate intracellular inflammatory signaling, e.g., peptide-gold nanoparticle hybrids that buffer endosomal pH have potently inhibited TLR-mediated inflammation in macrophages [60]. In addition, Muhammad et al. showed that ROS-responsive polymeric nanoparticles loaded with dexamethasone suppressed cytokine storms in acute lung injury by downregulating inflammatory pathways and reducing neutrophil infiltration [61]. Thus, nanomaterials such as dPGS not only block the initial ligand/receptor interaction at the cell surface but may also continue to impede pro-inflammatory signaling inside the cell. This dual extra-/intracellular mode of action is a unique feature that distinguishes dPGS from most conventional anti-inflammatory agents, which typically function in one domain. In summary, this study positions dPGS as a versatile anti-inflammatory nanostructure targeting the inflammatory HMGB1/RAGE axis. It compares favorably with other charged polymers and demonstrates efficacy in both innate immune cells and neural cell contexts. The capacity of dPGS to act on extra- and intracellular targets of inflammation is of great interest. Looking forward, in vivo studies will be crucial to evaluate the pharmacokinetics, biodistribution, and therapeutic index of dPGS, particularly in sepsis and intracranial tumor models.

## 4. Materials and Methods

### 4.1. Cloning of Human HMGB1 into a Modified pET-22b(+) Vector

The standard pET-22b(+) expression vector (Merck, Darmstadt, Germany) was modified by including a serine-glycine-serine (SGS) linker and two additional histidines upstream of the 6 × His-Tag, resulting in a more flexible and higher-affinity 8 × His-Tag at the C-terminus. By using the following primer pair: 5′-AGCGGCAGCCATCATCACCACCACCACCACCAC-3′ and 5′-CTCGAGTGCGGCCGCAAGCTTGTCGAC-3′ in a back-to-back primer strategy, the linker DNA was added to the vector. For the PCR, the Q5 High Fidelity 2× Master Mix from New England Biolabs (NEB, Frankfurt am Main, Germany) was applied according to the manufacturer’s instructions. PCR Thermocycle settings were: initial denaturation at 98 °C for 30 s, 35 cycles with 98 °C denaturation for 10 s, annealing at 50 °C for 30 s, extension at 72 °C for 30 s, and final extension at 72 °C for 120 s using the PCR 2720 Thermo Cycler (Thermo Fisher Scientific, Dreieich, Germany). The PCR product was run on a 1% agarose gel and visualized with SYBR Safe DNA Gel staining (Thermo Fisher Scientific, Dreieich, Germany), cut out, and purified with a Zymoclean Gel DNA Recovery Kit (Zymo Research, Freiburg im Breisgau, Germany). Next, 1 µL of the purified PCR product was treated with the KLD (Kinase, Ligase, *DpnI*) mix (NEB, Frankfurt am Main, Germany) as described in the manufacturer’s protocol. 5 µL of the KLD reaction mixture was directly used for the transformation of chemically competent *E. coli* DH5α cells (NEB, Frankfurt am Main, Germany). Plasmids of transformants were isolated from overnight cultures using the NucleoSpin plasmid kit (Macherey-Nagel, Düren, Germany). Sequence integrity of the obtained clones was confirmed by sequencing (Eurofins Genomics, Berlin, Germany). A plasmid encoding human HMGB1 (hHMGB1) was obtained from Origen (HMG1 (HMGB1) (NM_002128) Human Tagged ORF Clone, SKU RG205918, Herford, Germany) was used as a template for polymerase chain reaction (PCR) amplification with the following primer pair: 5′-CTTTAAGAAGGAGATATACATATGGGCAAAGGAGATCCTAAG-3′, 5′-GTGATGATGGCTGCCGCTCTCG ATCATCATCAT CATCTTC-3′. The resulting PCR product with a size of 645 bp was inserted into the modified pET-22b(+) vector by the exonuclease and ligation-independent cloning (ELIC) method, described by Koskela et.al. [62]. In brief, the modified pET-22b(+) vector was digested with the enzymes *Nde*I*/Xho*I (Thermo Fisher Scientific, Dreieich, Germany) to create homologous ends. After gel extraction and DNA cleanup, a 1:3 molar ratio of digested pET-22b(+) and hHMGB1 PCR product was used for ligation at room temperature for 15 minutes and finally transformed into *E. coli*. Insert integrity was confirmed by sequencing and restriction analysis.

### 4.2. Recombinant Expression of hHMGB1

For bacterial expression of hHMGB1, the *E. coli* BL21 (DE3) (New England Biolabs, Frankfurt am Main, Germany) strain was used. The expression procedure followed the protocol described by Bergeron et al. with some changes [63]. Freshly transformed *E. coli* BL21 (DE3) was used to inoculate 50 mL of low salt (86 mM NaCl) LB-amp medium with the addition of 0.4% (*v*/*v*) glucose and incubated overnight at 200 rpm, 37 °C. The next day, the expression culture started with an OD_600nm_ of 0.1 in a total volume of 1 L. At an optical density of 0.5, protein expression was induced by the addition of 1 mM isopropyl-β-D-thiogalactopyranoside (IPTG) for 4 h at 30 °C. Bacteria were harvested at 8200× *g* for 30 min, 4 °C, and the cell pellet was resuspended on ice in 10 mL lysis buffer containing 50 mM Tris-HCl, pH 8.0, 100 mM NaCl, 2.5 mM MgCl_2_, 10 mM imidazole. For lysis of the bacteria, one spatula tip of lysozyme (Roche Diagnostics, Basel, Switzerland), 50 U/mL Benzonase nuclease (Merck, Darmstadt, Germany), and 300 µL of 25 × EDTA-free protease inhibitor cocktail (final concentration 1×, Sigma-Aldrich, Darmstadt, Germany) was added and the sample incubated for 30 min, 37 °C, 180 rpm. Complete lysis of the bacteria was achieved by ultrasonication on ice (Branson Sonifier 250, Brookfield, CT, USA) and the following settings: 15% output, 30 s pulse, 1 min resting time, for 15 min. The bacterial cell debris was removed by centrifugation for 30 min, 20,000× *g*, 4 °C, and the obtained supernatant was used for further purification.

### 4.3. Affinity Chromatography

All purification steps were performed on an Äkta pure 25 protein purification system (Cytiva, Uppsala, Sweden) at 4 °C.

#### 4.3.1. Ni-NTA Chromatography

A 1 mL HisTrap FF column (Cytiva, Dreieich, Germany) was used for the first purification step. The settings for the HisTrap FF column were applied as described by the manufacturer. In brief, using a flow rate of 1 mL/min, the column was prewashed with 10 column volumes (CV) (10 mL) of MilliQ water and equilibrated with 10 CV lysis buffer. The soluble bacterial supernatant was concentrated to 1 mL with Amicon filter units with a cutoff of 3 kDa (Merck, Darmstadt, Germany) and injected into a 1 mL sample loop mounted to the Äkta system. After loading the sample onto the column, the latter was washed with 15 CV of lysis buffer. Elution of bound protein was performed with an imidazole gradient of 20 CV mixing buffers containing 50 mM Tris-HCl, pH 8.0, with one containing an additional 500 mM imidazole from 0% to 100% imidazole-containing buffer, starting at 0% and ending with 100% which equals 500 mM imidazole. Fractions of 1 mL were collected with an automated fraction collector (Cytiva, Uppsala, Sweden) during the whole purification process. The elution profile was monitored at distinct wavelengths of 280, 260, and 215 nm. Fractions were analyzed on a 15% SDS-PAGE and peak fractions containing HMGB1 were pooled, concentrated, and buffer exchanged to a 1 mL sample in 50 mM Tris-HCl, pH 8.0, for further purification.

#### 4.3.2. Heparin-Sepharose Chromatography

A 1 mL HiTrap Heparin Sepharose FF (Cytiva, Dreieich, Germany) was used as a second column to increase the purity of the HMGB1 sample. The settings for the HiTrap Heparin Sepharose FF column were applied as described by the manufacturer. The flow rate was set to 0.2 mL/min, the column was washed with 10 CV of MilliQ water and equilibrated with an additional 10 CV of 50 mM Tris-HCl buffer, pH 8.0. The 1 mL sample from the Ni-NTA purification step was applied, and the column was washed with 10 CV 50 mM Tris-HCl, pH 8.0. Elution of bound proteins was achieved with an increasing salt gradient of 20 CV up to 2 M NaCl (100%). Fractions of 0.2 mL were collected, and the elution was followed by absorbance measurements at 280, 260, and 215 nm. The eluate was analyzed on a 15% SDS-PAGE and peak fractions containing HMGB1 were pooled and concentrated with Amicon 3kDa cutoff filter units, while the buffer was exchanged to 50 mM Tris-HCl, 150 mM NaCl. The final protein concentration was measured by photometry at A_280nm_ (NanoDrop, Thermo Fisher Scientific, Dreieich, Germany), and the molarity was calculated by the following parameters (obtained from ProtParam [32]): molecular weight 26,464.37 Da and an extinction coefficient of 21,550 M^−1^cm^−1^.

### 4.4. Sodium Dodecyl Sulfate-Polyacrylamide Gel Electrophoresis (SDS-PAGE)

For visualization of the expressed hHMGB1, a 15% SDS-PAGE was prepared as described by Laemmli et al. and Groth et al. [64,65]. The 15 µL samples of the overnight culture, induced culture, insoluble, soluble, Ni-NTA, and heparin fractions were mixed with 5 µL 4 × Laemmli buffer (final 1×), supplemented with 5% β-mercaptoethanol, and subsequently heated to 95 °C for 5 min. Samples were loaded and the gels were run for the first 20 min, 15 mA, and an additional 45 min, 20 mA. Gels were stained with Instant Blue Coomassie Protein Stain ISB1L (Abcam, Heidelberg, Germany) for 20 min with gentle agitation. Prestained Protein Ladder 10 to 180 kDa (Thermo Fisher, Dreieich, Germany) was used for estimation of the obtained protein molecular weight. Gels were visualized with the ChemiDoc MP (BioRad, Stuttgart, Germany).

### 4.5. Synthesis and Characterization of Dendritic Polyglycerol Sulfate (dPGS) and Cy5 Labeled Derivatives

The starting material was a 10,000 Da dendritic polyglycerol (dPGOH) core, which was obtained by glycidol polymerization [66]. Preparation of dPGS was done according to Alban et al. and Türk et al. [67,68]. Labeling of dPGS with the red cyanine dye Cy5 was done according to Arenhoevel et al. and Gröger et al. [69,70]. The dPGS used in this study had a molecular weight of 15 kDa with a degree of sulfation of 80%, a ζ-potential of −32.6 ± 0.7 mV, and a hydrodynamic diameter of 6.6 ± 2.2 nm.

### 4.6. RAW 264.7 Macrophage Culture

The tumor mouse macrophage cell line RAW 264.7 (TIB-71, ATCC, Manassas, VA, USA) was used as a model for studying inflammatory processes in vitro. RAW 264.7 cells were cultured in 1 × Dulbecco’s Modified Eagle Medium (DMEM) supplemented with 4 g/L D-glucose/L-glutamine, no pyruvate (Gibco, Thermo Fisher Scientific, Dreieich, Germany), 1% Penicillin-Streptomycin (Thermo-Fisher Scientific, Dreieich, Germany), and 10% fetal bovine serum (FBS, Sigma-Aldrich, Darmstadt, Germany) (complete culture medium). Cells were sub-cultured twice a week in ratios between 1:3 and 1:8 in a 25 cm^2^ culture flask (Corning, Berlin, Germany) at 37 °C, 5% CO_2_. Sub-culturing was performed by removing culture medium and adding 10 mL 1 × Dulbecco’s Phosphate-Buffered Saline (DPBS, no CaCl_2_, no MgCl_2_, Thermo Fisher Scientific, Dreieich, Germany) for washing the cells. Detachment of the cells was done by adding 1 mL of 0.25% trypsin-EDTA (Thermo Fisher Scientific, Dreieich, Germany) and subsequent incubation for 10 min, 37 °C, 5% CO_2_. Complete detachment of adherent cells was done by using a cell scraper. Trypsin was neutralized by adding 10 mL of complete culture medium and centrifugation at 800 rpm, 5 min, room temperature. The medium was removed, and the cell pellet was resuspended in 10 mL of fresh complete culture medium. Counting of RAW 264.7 macrophages was done with the Luna FL Dual fluorescence cell counter (Logos Biosystems, Villeneuve-d’Ascq, France), and viability was determined by the addition of equal volumes of cells and 0.4% trypan blue (Thermo Fisher Scientific, Dreieich, Germany) in LUNA 2-channel cell counting slides (Logos Biosystems, Villeneuve-d’Ascq, France). Subcultures were grown between 2 × 10^4^ cells/mL and 3.2 × 10^5^ cells/mL. The cell viability was between 90–97%.

### 4.7. HMC3 Human Microglia Culture

Human HMC3 microglia cells (CRL-3304) were initially obtained from ATCC (Manassas, USA) and cultured in 1 × DMEM supplemented with 4.5 g/L D-Glucose and L-Glutamine, 110 mg/L sodium pyruvate (Gibco, Thermo Fisher Scientific, Dreieich, Germany), 5% FBS (Gibco, Thermo Fisher Scientific, Dreieich, Germany), 1% Penicillin-Streptomycin (Invitrogen, Thermo Fisher Scientific, Dreieich, Germany) (complete growth medium-HMC3) at 37 °C, 5% CO_2_. Cells were sub-cultured at 70–80% confluency in a 25 cm^2^ culture flask (Corning, Berlin, Germany). First, the medium was removed, and the cells were washed with 10 mL 1 × DPBS and detached by adding 1 mL 0.05% trypsin-EDTA (Invitrogen, Thermo Fisher Scientific, Dreieich, Germany) for 1–2 min at 37 °C, 5% CO_2_. Trypsin was neutralized by adding 10 mL of complete growth medium-HMC3 and centrifugation at 800 rpm, 5 min, room temperature. The medium was removed, and the cell pellet was resuspended in 10 mL fresh complete growth medium-HMC3. Cells were sub-cultured twice a week in ratios between 1:3 and 1:8.

### 4.8. PLA

RAW 264.7 macrophages were seeded onto coverslips, precoated with poly-L-lysine (Corning, Berlin, Germany), to 10,000 cells/coverslip in complete culture medium, 24 h, 37 °C, 5% CO_2_. Next, cells were washed once with 1 × DPBS (no CaCl_2_, no MgCl_2_, Thermo Fisher Scientific, Dreieich, Germany) and treated with 100 ng/mL lipopolysaccharide (LPS from *E. coli* O111:B4, 1 mg/mL ready-made solution, Sigma-Aldrich, Darmstadt, Germany) as stimulants, 50 nM dPGS as a potential inhibitor, and a combination of 100 ng/mL LPS + 50 nM dPGS in serum-free 1 × DMEM for 24 h, 37 °C, 5% CO_2_. The sample without treatment served as an untreated control. The next day, cells were washed once again with 1 × DPBS (no CaCl_2_, no MgCl_2_, Thermo Fisher Scientific, Dreieich, Germany) and fixed with 4% paraformaldehyde (PFA, Thermo Fisher Scientific, Dreieich, Germany) in DPBS for 10 min, room temperature, and further permeabilized with 0.1% Triton X-100 (Thermo Fisher Scientific, Dreieich, Germany) in DPBS for 10 min, room temperature. Blocking was done for 1 h, as described by the manufacturer for the Duolink In Situ Orange Starter Kit Goat/Rabbit (Millipore-Sigma, Darmstadt, Germany). Primary antibodies (goat anti-human RAGE, Biotechne, Wiesbaden, Germany, and rabbit anti-human HMGB1, Abcam, Heidelberg, Germany, diluted 1:200 and 1:500, respectively, in 1 x Antibody Diluent) were incubated overnight at 4 °C in a humidified chamber. The next steps were carried out according to the manufacturer’s protocol. Actin labeling was done with Phalloidin-iFluor 488 (Abcam, Heidelberg, Germany) in a 1:500 dilution in DPBS for 20 min, room temperature, in the dark. Nuclei labeling was achieved with DAPI by using the Duolink mounting medium and incubation for 15 min at room temperature. Samples were imaged using the Zeiss Axio Observer. Z1 microscope and ZEN blue (version: 3.7.97.04000) software (Carl Zeiss, Jena, Germany). For HMC3 microglia cells, 7000 cells/coverslip were seeded under the above-mentioned conditions. The following steps were done according to the procedure described for the RAW 264.7 macrophages. In contrast, as primary antibodies, the rabbit anti-human RAGE (Abcam, Toronto, ON, Canada, 1:500 dilution) and the mouse anti-human HMGB1 [1F3] (Abcam, Toronto, ON, Canada, 1:500 dilution) were used. The following steps were applied according to the instructions for the Duolink In Situ Red Starter Kit Mouse/Rabbit (Millipore-Sigma, Oakville, ON, Canada). Actin labeling was performed for 20 min with Phalloidin Alexa Fluor 488 (1:400 dilution, Thermo Fisher Scientific, Oakville, ON, Canada), and nuclei staining was done with Hoechst 33342 trihydrochloride (Thermo Fisher Scientific, Dreieich, Germany) diluted 1:10,000 in 1 × DPBS for 10 min at room temperature. Samples were imaged using a fluorescence microscope (Leica DMI4000 B, Richmond Hill, ON, Canada).

### 4.9. NF-κB Shuttling

RAW 264.7 macrophages were seeded onto coverslips, precoated with poly-L-lysine (Corning, Berlin, Germany), to 10,000 cells/coverslip in complete culture medium, 24 h, 37 °C, 5% CO_2_. Next, cells were washed once with 1 × DPBS (no CaCl_2_, no MgCl_2_, Thermo Fisher Scientific, Dreieich, Germany) and treated with 1 µg/mL HMGB1, 100 ng/mL LPS (as stimulants), 50 nM dPGS (as a potential inhibitor), 1 µg/mL HMGB1 + 50 nM dPGS, 100 ng/mL LPS + 50 nM dPGS in serum-free 1 × DMEM for 90 min, 37 °C, 5% CO_2_. The sample without treatment served as the untreated control. After the treatment, cells were washed one more time with 1 × DPBS (no CaCl_2_, no MgCl_2_, Thermo Fisher Scientific, Dreieich, Germany) and subsequently fixed with 4% PFA in PBS for 10 min, room temperature. Permeabilization was done with 0.1% Triton X-100, 10 min, room temperature. Blocking was achieved with 5% normal goat serum (Thermo Fisher Scientific, Dreieich, Germany), 2% bovine serum albumin (BSA, Sigma-Aldrich, Darmstadt, Germany) in PBS for 1 h, room temperature. A primary antibody against NF-κB p65 (D12E12) XP Rabbit mAb (Cell Signaling Technology, Danvers, MA, USA) was diluted 1:400 in 5% normal goat serum, 2% BSA, and incubated overnight at 4 °C. On the next day, cells were washed 3 × with 1 × DPBS, 5 min each, and subsequently incubated with a goat anti-rabbit IgG (H+L) Cross-adsorbed secondary antibody, FITC labeled conjugate (Thermo Fisher Scientific, Dreieich, Germany) diluted 1:750 in 5% normal goat serum, 2% BSA, and incubated for 1 h, room temperature, in the dark. After three additional wash steps with 1 × DPBS, 5 min each, the nucleus was labeled with Hoechst 33342 trihydrochloride, trihydrate (Thermo Fisher Scientific, Dreieich, Germany) 1:10,000 in 1 × PBS for 10 min, room temperature, in the dark. Cells were washed twice with 1 × DPBS, 5 min each, and coverslips were mounted into Epredia Immu-Mount mounting medium (Thermo Fisher Scientific, Dreieich, Germany). The mounting medium was solidified overnight at room temperature. The next day, images were taken with a Zeiss Axio Observer. Z1 microscope and ZEN blue (version: 3.7.97.04000) software.

### 4.10. Reverse Transcription (RT)-PCR

RAW 264.7 macrophages were seeded in 6 well-plates at 3 × 10^5^ cells/well in complete medium, at 37 °C, 5% CO_2_ for 24 h. The next day, cells were washed with 1 × DPBS and treated with either 1 µg/mL HMGB1 or 100 ng/mL LPS (as stimulus) and 50 nM dPGS (as potential inhibitor) in serum-free medium for 6 h. Medium only was used as the untreated control. Cells were harvested, and the RNA was extracted as indicated by the NucleoSpin RNA Mini Kit instructions (Macherey-Nagel, Düren, Germany). RNA concentration and purity were determined at 260 nm with NanoDrop (Thermo Fisher Scientific, Dreieich, Germany). The reverse-transcription PCR was performed with the One-Step RT-PCR Kit (Qiagen, Hilden, Germany) by using 250 ng of purified RNA. Primer sequences used are listed in Table 1. Annealing temperatures were set to 56 °C for TNF-α and β-actin (ACTB, a housekeeping gene) and 59 °C for IL-6 and MCP-1. Analysis of the RT-PCR results was done by a 2% agarose gel, and bands were visualized as described above. Band intensities were analyzed with Fiji ImageJ (version 1.54). The Gene Ruler 100 bp DNA Ladder (Thermo Fisher Scientific, Dreieich, Germany) was used to identify the correct PCR product size.

### 4.11. Quantitative RT-PCR

RAW 264.7 cells (generously gifted to PGW and SKS by Prof. R.E.W. Hancock, UBC) were seeded in a 6-well plate (1 × 10^6^ cells/well) and cultured overnight at 37 °C in complete culture medium in a CO_2_ incubator. The cells were then co-stimulated with LPS from *E. coli* at 100 ng/mL in the presence of dPGS (1.6 µM). Stimulated cells were incubated for 48 h at 37 °C in a CO_2_ incubator. Total RNA was extracted from the cells using a RNeasy kit (Qiagen, Toronto, ON, Canada) according to the manufacturer’s instructions. The integrity of the extracted RNA was determined at 260 nm using a NanoDrop (Thermo Fisher Scientific, Mississauga, ON, Canada) device. Reverse transcription was performed using iScript select cDNA synthesis kit (Bio-Rad, Mississauga, ON, Canada) with 100 ng RNA and 1 μL oligo(dT)18. Real-time PCR was performed in the CFX Opus Real-Time PCR system (Bio-Rad, Mississauga, ON, Canada) using a reaction mixture that consists of SSO advance Syber green super mix (Bio-Rad, Mississauga, ON, Canada), cDNA template, forward and reverse primers (see Table 1). The relative expression of the target genes was calculated using the 2^−ΔΔCt^ method using β-actin as the reference gene. All experiments were repeated using three biological replicates, with two technical replicates each.

### 4.12. NO Assay

The accumulated NO produced by the cell culture was measured in the form of nitrite using a Griess reagent kit (Promega, Madison, WI, USA). RAW 264.7 cells (generously gifted to PGW and SKS by Prof. R.E.W. Hancock, UBC) were plated at a density of 1 × 10^5^ cells/well and were allowed to adhere overnight at 37 °C in a humidified atmosphere containing 5% CO_2_. After incubation, the cell culture medium was replaced with fresh medium containing 100 ng/mL of LPS from *E. coli* and dPGS (0.0, 0.2, 0.4, 0.8, 1.6, 3.2, 6.4, 12.8, 25.6 µM), followed by incubation at 37 °C for 24 h in a CO_2_ incubator. Cells not treated with dPGS and LPS were used as the untreated control. Accumulated nitrite in the culture supernatant after 24 h was measured by using the Griess reagent. 50 mL of culture supernatant was mixed with an equal volume of 1% sulfanilamide in 5% phosphoric acid. After incubation for 10 min at room temperature, 50 μL of 0.1% N-1-naphthylethylenediamine dihydrochloride (NED) was added and incubated for another 20 min at room temperature. The absorbance was measured at 570 nm using a microplate reader. Nitrite concentrations were determined using a sodium nitrite calibration curve. The error is reported as the standard deviation obtained from 3 biological replicates for each condition.

### 4.13. ELISA

Cells were seeded into a 96-well plate at 10,000 cells/well in complete culture medium, 37 °C, 5% CO_2_, and grown for 24 h. On the next day, cells were washed once with 1 × DPBS and subsequently treated with either 1 µg/mL HMGB1 or 100 ng/mL LPS and the indicated dPGS concentrations (0.05 to 500 nM) in serum-free conditions at 37 °C, 5% CO_2_ for 16 h. At the end of the treatment, the cells were centrifuged at 800 rpm, 5 min, room temperature, the supernatant was targeted for analyses by ELISA, and the cells were used for MTT-assay. For quantification of secreted cytokines, MCP-1, IL-6, and TNF-α, the mouse DuoSet Kits and DuoSet Ancillary Reagent Kit 2 from Bio-Techne (Wiesbaden, Germany) were used. The ELISA was performed following the manufacturer’s protocol. The photometric readout at 450 nm was done with the Infinite M200 plate reader (Tecan Magellan, Crailsheim, Germany). Analyses of the data were done by blank referencing and using the 4-parameter logistic progression fit within the Origen 2021 software (version 9.8.0.200).

### 4.14. MTT-Assay

Cells were seeded and treated as described for the ELISA. After the treatment, cells were incubated with 100 µL 0.5 mg/mL MTT (3-(4,5-dimethylthiazol-2-yl)-2,5-diphenyltetrazolium bromide) from Merck (Darmstadt, Germany), diluted in 1 × DMEM at 37 °C for 1 h. The medium was removed, and formazan crystals were dissolved in dimethyl sulfoxide (DMSO, Merck, Darmstadt, Germany). Colorimetric measurement was done at 595 nm with the Infinite M200 plate reader (Tecan Magellan, Crailsheim, Germany).

### 4.15. Time-Dependent dPGS-Cy5 Uptake Studies

#### 4.15.1. Uptake of dPGS-Cy5 into RAW 264.7 Macrophages

To analyze the time-dependent uptake of dPGS-Cy5, RAW 264.7 macrophages were seeded in 24-well plates at a density of 1 × 10^5^ cells/well in 500 μL serum-containing DMEM (Thermo Fisher Scientific, Waltham, MA, USA), supplemented with 10% FBS (Thermo Fisher Scientific, Waltham, MA, USA) and 1% P/S (Sigma-Aldrich, St. Louis, MO, USA) and allowed to adhere at 37 °C and 5% CO_2_. Cells were washed once with DPBS (Thermo Fisher Scientific, Waltham, MA, USA) and subsequently incubated with 50 nM dPGS-Cy5 in serum-free DMEM for the indicated time points (1, 1.5, 6, 16, and 24 h). Next, the cells were washed twice with DPBS and detached by incubation with 250 μL 0.05% trypsin-EDTA (Thermo Fisher Scientific, Waltham, MA, USA) for 20 min at 37 °C and 5% CO_2_. The resuspended cell suspension in 250 µL complete DMEM was transferred to 1.5 mL Eppendorf tubes and centrifuged at 140 × *g* at 4 °C for 4 min. The cell pellet was resuspended in 300 μL flow cytometry buffer (2% BSA (Carl Roth GmbH, Karlsruhe, Germany), 2.5 mM EDTA (Carl Roth GmbH, Karlsruhe, Germany) in DPBS pH 7.4) on ice. The flow cytometry measurements were performed using an Attune NxT Acoustic Focusing Cytometer (Thermo Fisher Scientific, Waltham, MA, USA). The sample flow rate was set to200 μL/min, and the voltages of the forward and side scatter lasers (FSC and SSC) were set to 100 and 310 V, respectively. dPGS-Cy5 was detected in the RL1 channel (ex. 638 nm at 300 V, 670 ± 14 nm band pass filter). A total of 10,000 single cells per sample were analyzed. Data analysis was performed with FlowJo 10.10.0 software. Live cells were gated based on the side and forward scatter area (SSC-A/FSC-A), followed by the selection of single cells based on a linear relationship between the forward scatter height and area (FSC-H/FSC-A). The cellular uptake of dPGS-Cy5 was quantified as MFI. The background fluorescence of cells was removed by subtracting the MFI of non-fluorescent controls. The Cy5-MFI values were then normalized to the dPGS-Cy5 unstimulated control of the respective time point. The experiment was independently repeated three times, each time with at least three technical replicates.

#### 4.15.2. Uptake of dPGS-Cy5 into Macrophages upon Co-Stimulation with LPS or HMGB1

To analyze the uptake of dPGS-Cy5 into RAW 264.7 macrophages upon co-stimulation, the cells were seeded as described above. Cells were washed once with DPBS (Thermo Fisher Scientific, Waltham, MA, USA) and subsequently stimulated with 100 ng/mL LPS (Sigma-Aldrich, St. Louis, MO, USA) or 1 μg/mL recombinant HMGB1 in the presence of 50 nM dPGS-Cy5 in serum-free DMEM. The 50 nM dPGS-Cy5 in serum-free medium alone and only serum-free medium were used as unstimulated and non-fluorescent control, respectively. After indicated timepoints (1, 6, 16, and 24 h), the cells were washed twice, and sample preparation as well as flow cytometry data acquisition were performed and analyzed as previously described. The experiment was independently repeated three times, each time with at least three technical replicates.

### 4.16. Continuous Wave Electron Paramagnetic Resonance (Cw-EPR) Spectroscopy

The recombinant Human RAGE Protein (His-Tag, cat.# 11629-H08H, Sino Biological, Eschborn, Germany) was reconstituted by adding 500 µL Milli-Q water to the vial. Subsequently, it was concentrated to 50 µL using Amicon^®^ Ultra centrifugal filter units (3 kDa molecular weight cutoff, total volume 500 µL, Merck Millipore, Darmstadt, Germany). For HMGB1 spin-labeling, a 20 mM MTSL (TRC, North York, ON, Canada) stock solution in acetonitrile was used. To attach the label to the protein, HMGB1 and MTSL were incubated at 4 °C at an equimolar amount for 24 h. To remove unreacted spin label, Amicon^®^ Ultra centrifugal filter units (3 kDa molecular weight cutoff, total volume 500 µL; Merck Millipore, Darmstadt, Germany) were used. The sample was centrifuged (Centrifuge 5425 R, Eppendorf, Wesseling-Berzdorf, Germany) 2 times at 14.000 rpm for 30 min at 4 °C, while simultaneously performing a buffer exchange to the EPR-buffer (150 mM NaCl, 25 mM HEPES, pH = 7.0). Cw-EPR measurements were performed using a Bruker BR420 X-Band spectrometer upgraded with a Bruker ECS 041 xG microwave bridge and a lock-in amplifier (Bruker ER023M, Billerica, MA, USA). A 200 mT wide spectra were taken using a customized aqueous EPR flat cell (Hellma, Mühllheim, Germany), positioned in a Bruker ER422 SHQ 8304 (SHQ) resonator. The measurements were performed using a modulation frequency of 100 kHz, a modulation amplitude of 3 G, and a 5 mW power attenuation of 20 dB. For all measurements, the time constant was set to 20 ms, while the conversion time was set to 80 ms.

### 4.17. Statistics

Data are presented as the mean and standard deviation (SD) or standard error of the mean (SEM). Statistical significance was determined using one-way or two-way analysis of variance (ANOVA), followed by a multiple comparison test (Tukey’s, Dunnett’s, or Tukey-Kramer post hoc). The *p*-values less than 0.05 were considered significant. Experiments were repeated independently at least three times, unless otherwise stated. Statistical analysis was performed using GraphPad Prism (version 10.5.0).

## Figures and Tables

**Figure 1 ijms-26-10440-f001:**
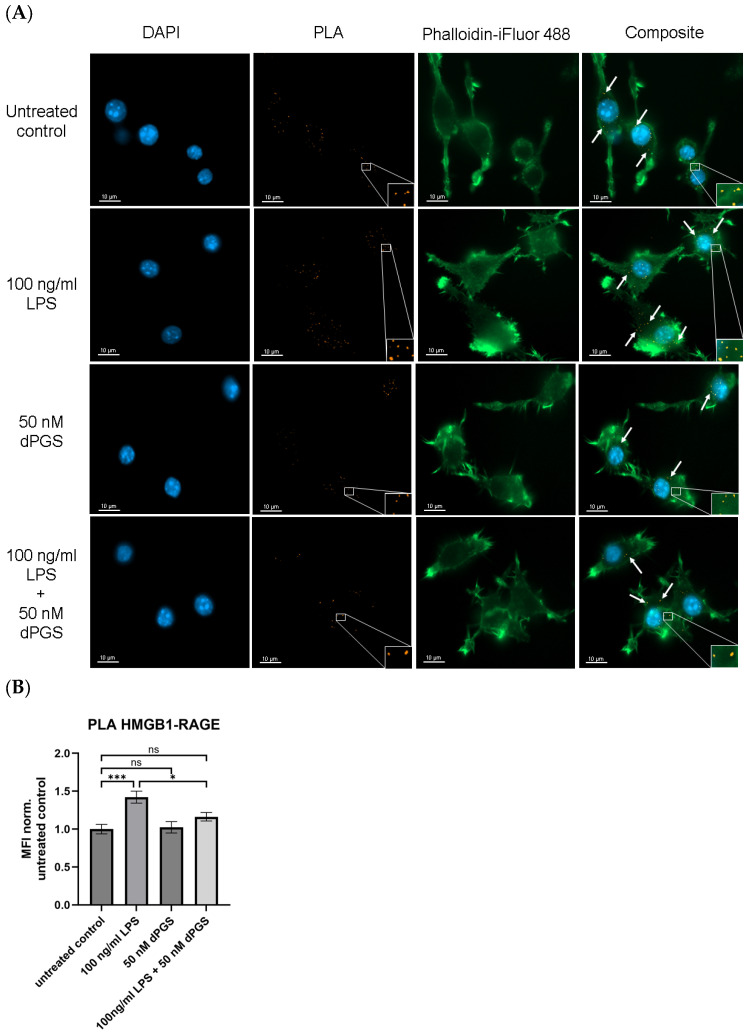
dPGS reduces HMGB1/RAGE interactions in LPS-stimulated RAW 264.7 macrophages. (**A**) RAW 264.7 macrophages were seeded onto glass coverslips at 10,000 cells/coverslip. Cells were then treated with 100 ng/mL LPS, 50 nM dPGS, and 100 ng/mL LPS + 50 nM dPGS under serum-free conditions for 24 h. PLA visualized the interaction of HMGB1 and RAGE, as indicated by orange dots and white arrows. Phalloidin-iFluor 488 (green) and DAPI (blue) were used to visualize actin and nuclei, respectively. The scale bars represent 10 µm, and images were taken with a Zeiss Axio Observer. Z1 microscope and the ZEN blue software (version: 3.7.97.04000). (**B**) Evaluation of data obtained by PLA. The mean fluorescence intensity (MFI) per image of the PLA signal was normalized to that of the untreated control and set to 1. Bar graphs show the mean values ± SEM. Statistical analysis was assessed using two-way ANOVA, followed by Tukey’s multiple comparison test. At least 40 cells in three independent experiments were analyzed (*** *p* ≤ 0.001, * *p* ≤ 0.05, ns = not significant). The diagram was generated with GraphPad Prism 10 (version 10.5.0).

**Figure 2 ijms-26-10440-f002:**
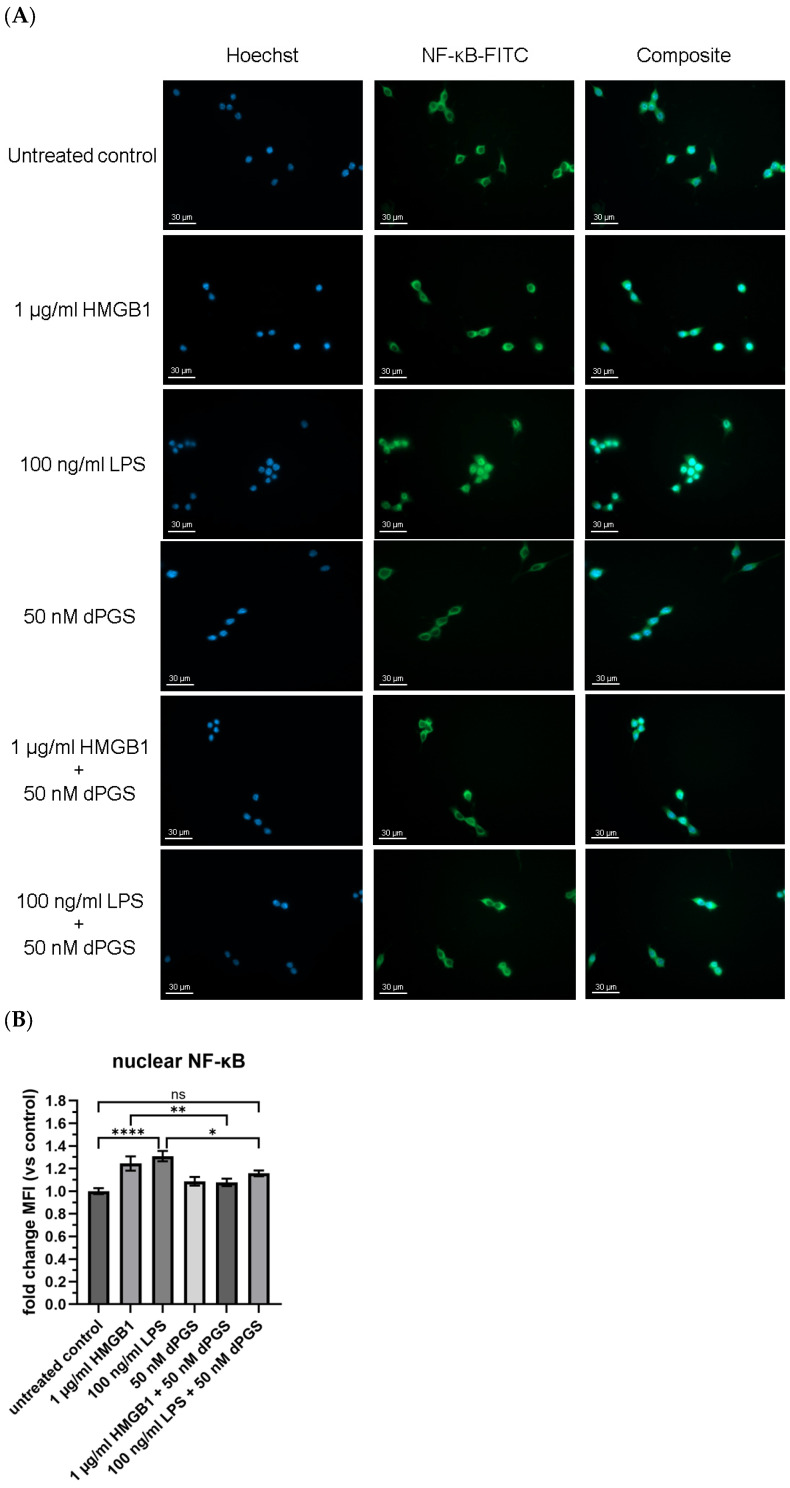
Shuttling of NF-κB is inhibited by dPGS. (**A**) RAW 264.7 macrophages were seeded onto glass coverslips at 10,000 cells/coverslip. Cells were then treated with 1 µg/mL HMGB1, 100 ng/mL LPS, 50 nM dPGS, and combinations under serum-free conditions for 90 min. For visualization of NF-κB, cells were incubated with an NF-κB p65 rabbit mAb followed by an anti-rabbit FITC-conjugated antibody. The nucleus was labeled with Hoechst 33342, and images were taken with a Zeiss Axio Observer. Z1 microscope and the ZEN blue (version: 3.7.97.04000) software. (**B**) The nuclear MFI was calculated for each sample and normalized to the untreated control (set to 1). At least 70 cells in three independent experiments were analyzed. Statistical analysis was performed by using two-way ANOVA followed by Tukey’s multiple comparisons test. Mean ± SEM are shown (**** *p* ≤ 0.0001, ** *p* ≤ 0.01, * *p* ≤ 0.05, ns = not significant). Scale bar 30 µm. The diagram was generated with GraphPad Prism (version 10.5.0).

**Figure 3 ijms-26-10440-f003:**
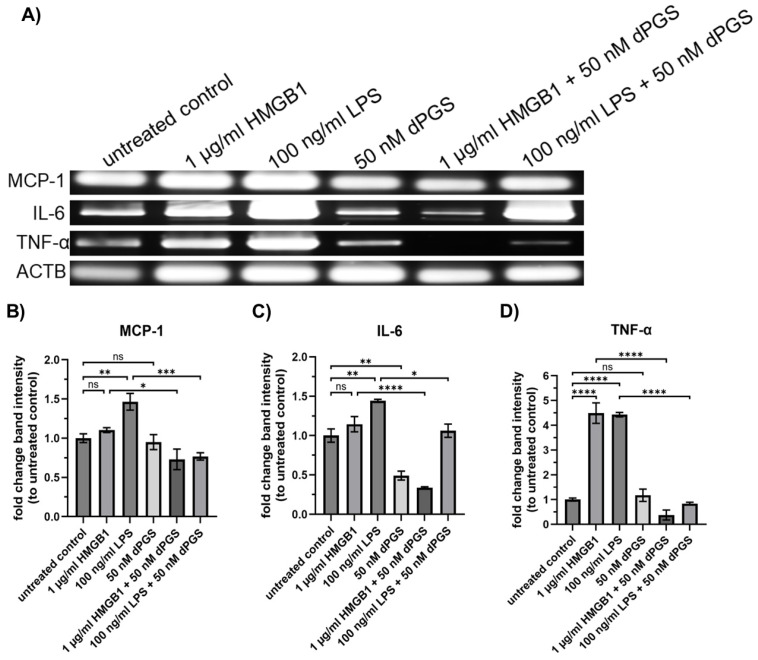
dPGS reduces mRNA levels of HMGB1-stimulated cytokines. (**A**) Gel-separated RT-PCR products received after indicated treatments of RAW 264.7 macrophages. (**B**–**D**) Quantification of cytokines was performed using ImageJ (version 1.54); values were normalized to the untreated control. Respective bands for MCP-1 (271 bp), IL-6 (474 bp), TNF-α (795 bp), and ACTB (150 bp) used as house-keeping gene. At least three independent experiments were performed. Statistical analysis was done by using one-way ANOVA followed by Tukey’s multiple comparison test. Mean ± SEM are shown (**** *p* ≤ 0.0001, *** *p* ≤ 0.001, ** *p* ≤ 0.01, * *p* ≤ 0.05, ns = not significant). The diagrams were generated with GraphPad Prism 10 (version 10.5.0).

**Figure 4 ijms-26-10440-f004:**
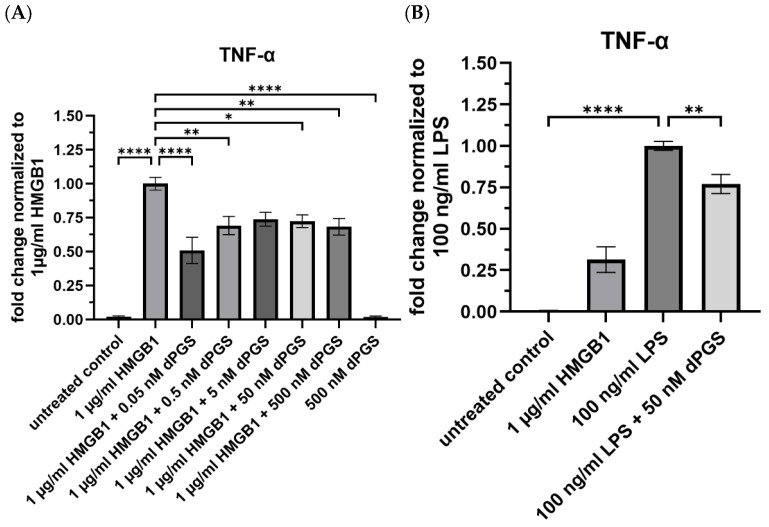
Release of pro-inflammatory TNF-α from stimulated RAW 264.7 macrophages is reduced by dPGS. RAW 264.7 macrophages were seeded at 10,000 cells/well in a 96-well plate in complete medium and stimulated with respective treatments, (**A**) 1 µg/mL HMGB1 +/− 500 nM dPGS, 50 nM dPGS, 5 nM dPGS, 0.5 nM dPGS, 0.05 nM dPGS, and (**B**) 100 ng/mL LPS +/− 50 nM dPGS in serum-free medium for 16 h. Medium alone was used as an untreated control, and 500 nM dPGS as a polymer control. Secreted TNF-α was determined by ELISA and normalized to the untreated control. Data are shown as (**A**) fold change relative to 1 µg/mL HMGB1 and (**B**) relative to 100 ng/mL LPS treatment. At least three independent experiments were performed. The mean ± SEM are shown. Statistical analysis was done by using two-way ANOVA followed by Dunnett’s multiple comparison test (**** *p* ≤ 0.0001, ** *p* ≤ 0.01, * *p* ≤ 0.05). The diagrams were generated with GraphPad Prism 10 (version 10.5.0).

**Figure 5 ijms-26-10440-f005:**
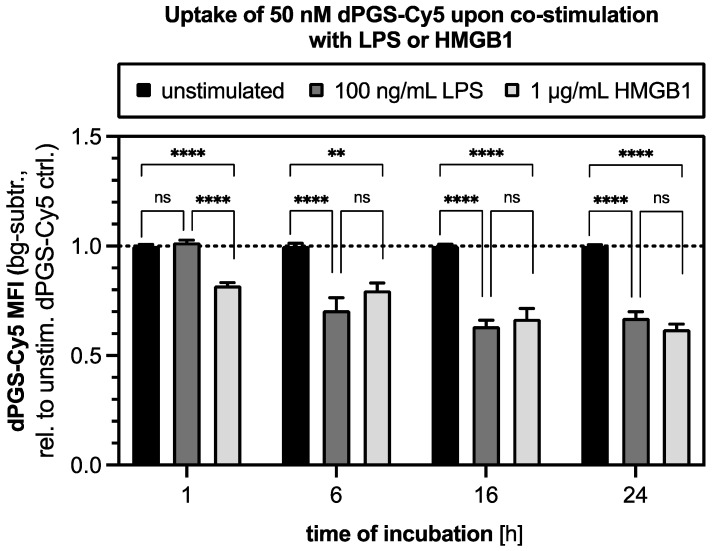
Co-incubation with LPS or HMGB1 reduces the uptake of dPGS-Cy5 into RAW 264.7 macrophages. In total, 1 × 10^5^ cells/well were seeded in 24-well plates and incubated with 50 nM dPGS-Cy5 in serum-free medium, without stimulus or in the presence of 1 μg/mL HMGB1 or 100 ng/mL for 1, 6, 16, and 24 h. The cellular uptake of dPGS-Cy5 was analyzed by flow cytometry and quantified as a fold change in Cy5 MFI relative to the unstimulated dPGS-Cy5 control (shown by a dotted line), after subtraction of the non-fluorescent background control. The data are shown as the mean ± SEM from three independent experiments, each performed with at least three technical replicates. For each time point, statistical significance was determined by one-way ANOVA, followed by Tukey-Kramer post hoc testing. **** *p* ≤ 0.0001, ** *p* ≤ 0.01, ns *p* > 0.05. bg-subtr. = background-subtracted; unstim. = unstimulated.

**Figure 6 ijms-26-10440-f006:**
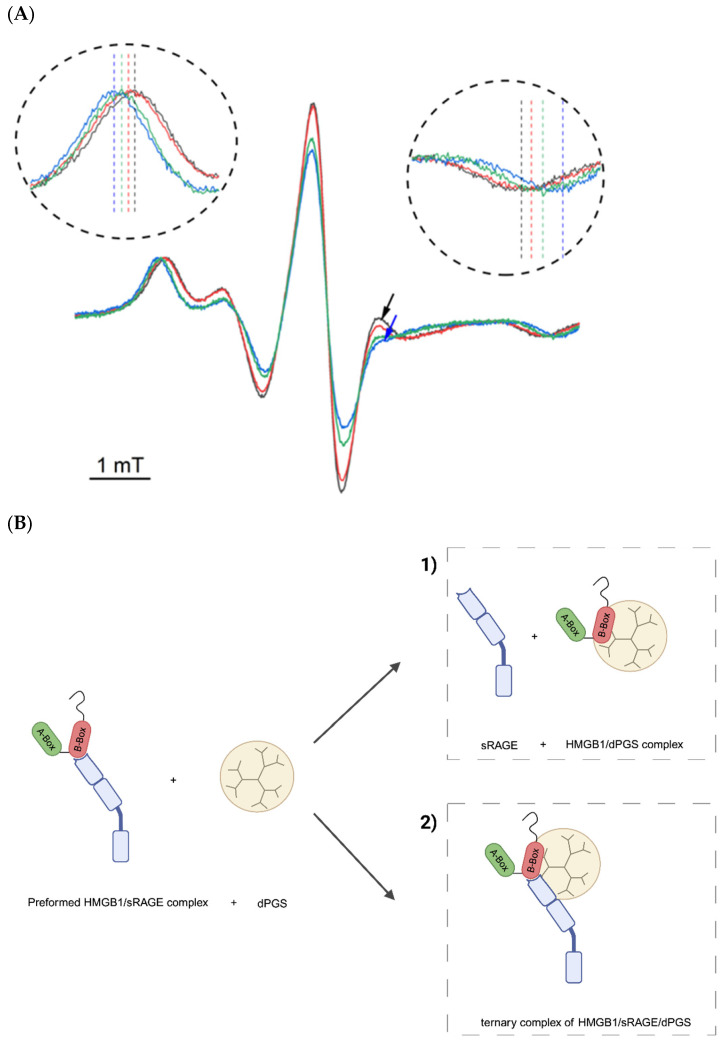
Binding analysis of HMGB1, sRAGE, and dPGS by electron paramagnetic resonance (EPR) spectroscopy and schematic illustration of potential complex formations. (**A**) EPR spectra of HMGB1 (black trace, 2A_zz_ = 61 G), HMGB1 + sRAGE premixed with 1 vol.% P20 (red trace, molar ratio 6:1, 2A_zz_ = 62.5 G), 11 µM HMGB1 + 2 µM sRAGE premixed with 1 vol.% P20 + 4 µM dPGS (blue trace, molar ratio 6:1:2, 2A_zz_ = 65.5 G), 20 µM HMGB1 + 20 µM dPGS (green trace, molar ratio 1:1, 2A_zz_ = 64.0 G). (**B**) Schematic representation of two different interaction scenarios upon addition of dPGS to the preformed HMGB1/sRAGE complex. Figure created with BioRender.com. (1) dPGS binds competitively to HMGB1, displacing sRAGE, occupying similar or overlapping binding sites, or (2) dPGS forms a ternary complex with both HMGB1 and sRAGE, hence, different binding sites.

**Table 1 ijms-26-10440-t001:** Forward and reverse primers used for RT-qPCR.

Gene	Forward Primer (5′-3′)	Reverse Primer (5′-3′)
*TNF-α*	GTTCTGTCCCTTTCACT CACTG	GGTAGA GAATGGATGAACACC
*iNOS*	CTGCAGCACTTGGATCA GAACCTG	GGGA GTAGCCTGTGTGCACCTGGAA
*IL-6*	CATGTTCTCTG GAAATCGTGG	AACGCA CTAGGTT TGCCGAGTA
*COX-2*	TTTGCCCAGCACTTCACCCAT	AAGTGGTAAC CGCTCAGGTGT
*MCP-1*	TGATCCCAATGAGTAGGCTG GAG	ATGTCTG GACCCATTCTTTCTTG
*β* *-actin*	GTGGGCCGCCCTAGGACCAG	GGAGGAAGAG GATGCGGCAGT

## Data Availability

The original data presented in the study are openly available in Zenodo at 10.5281/zenodo.17045830.

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
