# Peer review of "Dendritic Polyglycerol Sulfate Reduces Inflammation Through Inhibition of the HMGB1/RAGE Axis in RAW 264.7 Macrophages"

_ijms, 2025, doi:10.3390/ijms262110440_

Round 1

Reviewer 1 Report

Comments and Suggestions for Authors

The authors submitted a manuscript titled "Dendritic Polyglycerol Sulfate Reduces Inflammation through Inhibition of the HMGB1/RAGE Axis in RAW 264.7 Mouse Macrophages" as a review to the journal.
In this manuscript, Marten Kagelmacher et al. investigate the ability of dendritic polyglycerol sulfate (dPGS), a hyperbranched polyanionic polymer, to modulate High Mobility Group Box 1 (HMGB1)-induced signaling in RAW 264.7 mouse macrophages and human microglia.
This manuscript is well written.
I recommend producing a graphical abstract, also using software.
Please provide a better justification for the choice of dPGS concentrations.
I recommend enriching the discussion with other recent citations.

Author Response

Comment: I recommend producing a graphical abstract, also using software.

Response: We thank the reviewer for the valuable suggestion. A graphical abstract has been created using BioRender software and is now included in the resubmitted manuscript.

Comment: Please provide a better justification for the choice of dPGS concentrations.

Response: Thank you very much for this very important question. The range of dPGS concentrations (0.05 nM to 500 nM in cell-based assays and up to micromolar levels in biochemical assays) was chosen based on several considerations. First, previous reports demonstrated that dPGS exerts potent anti-inflammatory and protein-binding effects at low nanomolar to micromolar concentration, depending on the experimental system and molecular target [e.g., [1-4]]. To ensure that both sub-effective and saturating concentrations were included, we applied a series of dPGS, covering several orders of magnitude. Second, the 50 nM concentration was selected as a representative working dose for most cellular experiments (e.g., PLA, NF-κB translocation, RT-PCR, and uptake assays) because it (i) showed reproducible inhibition of HMGB1- and LPS-induced signaling, (ii) did not alter basal cell morphology or viability as confirmed by MTT assays (Figure S6), and (iii) provided a physiologically relevant balance between inhibitory efficacy and absence of cytotoxic or unspecific effects. Third, higher micromolar concentrations were only used in complementary biochemical assays (e.g., NO-release) to explore potential dose-dependent effects in pathways that require stronger suppression. Overall, the selected dPGS concentration range revealed first empirical data. A further adjustment is planned to optimize the treatment.

References for response 2

  1. Türk, H., R. Haag, and S. Alban, Dendritic Polyglycerol Sulfates as New Heparin Analogues and Potent Inhibitors of the Complement System. Bioconjugate Chemistry, 2004. 15(1): p. 162-167.
  2. Dernedde, J., et al., Dendritic polyglycerol sulfates as multivalent inhibitors of inflammation. Proc Natl Acad Sci U S A, 2010. 107(46): p. 19679-84.
  3. Maysinger, D., et al., Sulfated Hyperbranched and Linear Polyglycerols Modulate HMGB1 and Morphological Plasticity in Neural Cells. ACS Chem Neurosci, 2023. 14(4): p. 677-688.
  4. Joma, N., et al., Charged dendrimers reduce glioblastoma viability by modulating lysosomal activity and HMG1-RAGE interaction. Biochemical Pharmacology, 2025: p. 116969.

Comment: I recommend enriching the discussion with other recent citations.

Response: The discussion section was expanded by the published work shown in the manuscript reference numbers 47, 48, 54, and 60.

Reviewer 2 Report

Comments and Suggestions for Authors

Authors discovered that dPGS restricted HMGB1-RAGE ineraction, nuclear translocation of NF-κB and LPS-induced transcription of TNF-α, IL-6, MCP-1, COX-2, iNOS, and TNF-α secretion and nitric oxide production. It’s interesting. I have the following comments.

  1. Lots of discussions are in the results. Putting these content in the discussion may be better.
  2. It’s known that RAW264.7 cells are mouse macrophages. There is no need to emphasize these are mouse cells.
  3. In result 2.2 HMGB1/RAGE interaction is reduced by dPGS, add co-IP to prove HMGB1/RAGE interaction.
  4. In result 2.3. ShuÄ´ling of NF-κB in HMGB1 and LPS stimulated macrophages is reduced by dPGS, detect the protein levels of NF-κB in nucleus/cytoplasm fractionation by WB.
  5. In result 2.4. Dendritic polyglycerol sulfate reduces mRNA levels of cytokines, use qPCR qPCR to detect the mRNA levels of MCP-1, IL-6 and TNF-α.
  6. There is no need to describe the methods in figure legends.

Author Response

Comment: Lots of discussions are in the results. Putting this content in the discussion may be better.

Response: We thank the reviewer for this valuable suggestion. We have carefully revised the manuscript and moved several interpretative and explanatory passages from the Results section to the Discussion to ensure a clearer separation between data presentation and interpretation.

Comment: It’s known that RAW264.7 cells are mouse macrophages. There is no need to emphasize these are mouse cells.

Response: We thank the reviewer for this helpful comment. We have carefully revised the manuscript and substantially reduced the use of the word “mouse” throughout the text, including in the title. The term is now only retained where it is necessary for clarity.

Comment: In result 2.2 HMGB1/RAGE interaction is reduced by dPGS, add co-IP to prove HMGB1/RAGE interaction.

Response: Thank you for your suggestion. We agree that co-immunoprecipitation (co-IP) is a classical approach to confirm protein–protein interactions. However, the interaction between HMGB1 and RAGE has been comprehensively characterized and validated in multiple independent studies using co-IP, SPR, and structural methods. For instance, Hori et al. first demonstrated a direct HMGB1/RAGE association [1], and subsequent work by Park et al. confirmed this binding using recombinant proteins and RAGE-expressing macrophages [2]. In our study, the Proximity Ligation Assay (PLA) provides a robust, in situ method to detect HMGB1/RAGE interactions at
single-molecule resolution within intact cells. Moreover, our PLA data clearly demonstrate increased HMGB1/RAGE complex formation upon LPS stimulation and its disruption by dPGS, consistent with the well-established biology of this receptor–ligand pair. Therefore, we believe that an additional co-IP experiment would be redundant.

Resources to answer the comment

  1. Hori, O., et al., The receptor for advanced glycation end products (RAGE) is a cellular binding site for amphoterin. Mediation of neurite outgrowth and co-expression of rage and amphoterin in the developing nervous system. J Biol Chem, 1995. 270(43): p. 25752-61.
  2. Park, J.S., et al., High mobility group box 1 protein interacts with multiple Toll-like receptors. Am J Physiol Cell Physiol, 2006. 290(3): p. C917-24.

Comment: In result 2.3. Shuttling of NF-κB in HMGB1 and LPS stimulated macrophages is reduced by dPGS, detect the protein levels of NF-κB in nucleus/cytoplasm fractionation by WB.

Response: We fully agree with the reviewer that nuclear/cytoplasmic fractionation followed by Western blotting would provide valuable quantitative confirmation of
NF-κB translocation. Indeed, such experiments are an excellent complement to immunofluorescence imaging and will be included in our ongoing follow-up studies focusing on the molecular mechanisms of dPGS-mediated NF-κB regulation. In the present work, our goal was to provide a first indication that dPGS interferes with NF-κB activation in HMGB1- and LPS-stimulated macrophages. The immunofluorescence analysis shown here reliably visualizes p65 localization changes at the single-cell level and therefore offers a valid qualitative insight into the inhibitory effect of dPGS on NF-κB nuclear translocation. We agree that Western blotting of nuclear and cytoplasmic fractions will further substantiate these findings and plan to include such data in a forthcoming publication.

Comment: In result 2.4. Dendritic polyglycerol sulfate reduces mRNA levels of cytokines, use qPCR qPCR to detect the mRNA levels of MCP-1, IL-6 and TNF-α.

Response: The qPCR data for MCP-1, IL-6, TNF-α, iNOS, and COX-2 are already included in the Supplementary Figure S4. of the manuscript. These results demonstrate that dPGS significantly reduces the mRNA expression levels of the indicated inflammatory markers in LPS-stimulated macrophages. If the reviewer has any further questions or comments regarding these data or their presentation, we would be very happy to provide additional clarification.

Comment: There is no need to describe the methods in figure legends.

Response: Thanks for this comment. The descriptions of experimental methods in the figure legends have been substantially reduced. Only essential details were retained to ensure proper understanding of the figures, while the full descriptions are provided in the methods section.

Round 2

Reviewer 2 Report

Comments and Suggestions for Authors

Pleae revise the manuscript according to the following comments which have been mentioned before.

In result 2.2 HMGB1/RAGE interaction is reduced by dPGS, add co-IP to prove HMGB1/RAGE interaction.

In result 2.3. Shuttling of NF-κB in HMGB1 and LPS stimulated macrophages is reduced by dPGS, detect the protein levels of NF-κB in nucleus/cytoplasm fractionation by WB.

Author Response

Comments In result 2.2 HMGB1/RAGE interaction is reduced by dPGS, add co-IP to prove HMGB1/RAGE interaction. (And) In result 2.3. Shuttling of NF-κB in HMGB1 and LPS stimulated macrophages is reduced by dPGS, detect the protein levels of NF-κB in nucleus/cytoplasm fractionation by WB.

Response to both comments: 

We thank the reviewer for their thorough evaluation of our work. However, as already noted in our previous response, we respectfully consider that the additional experiments suggested would not provide further information to substantiate the scientific conclusions of this study. The techniques proposed have been extensively applied in earlier investigations addressing similar questions, and the results of those studies are fully consistent with our findings. We therefore regard the conclusions presented in this manuscript as well-supported by the comprehensive experimental evidence already provided. Without a clear scientific rationale specifying which aspects of our data are considered insufficient, the additional scientific value of the requested experiments remains unclear to us, and we have therefore not performed them.

Regarding the specific suggestion to perform co-immunoprecipitation (Co-IP) experiments, we would like to point out that, while Co-IP is indeed useful for detecting protein–protein interactions, it is not ideally suited for the purpose of our study. First, Co-IP requires cell lysis, which disrupts subcellular compartmentalization and may lead to artificial or non-physiological interactions as well as non-specific binding to the immunoprecipitation antibody. This can compromise specificity and physiological relevance. Second, Co-IP is generally a qualitative technique and does not readily provide quantitative information on interaction frequency or spatial localization within intact cells. In contrast, the proximity ligation assay (PLA) we employed offers both qualitative and quantitative data at the single-cell level and allows visualization of interactions in situ under near-physiological conditions. For these reasons, PLA represents the more appropriate and informative method for our experimental objectives.